# Decreasing Incidence and Mortality in Traumatic Brain Injury in Korea, 2008–2017: A Population-Based Longitudinal Study

**DOI:** 10.3390/ijerph17176197

**Published:** 2020-08-26

**Authors:** Han-Kyoul Kim, Ja-Ho Leigh, Ye Seol Lee, Yoonjeong Choi, Yoon Kim, Jeong Eun Kim, Won-Sang Cho, Han Gil Seo, Byung-Mo Oh

**Affiliations:** 1Department of Rehabilitation Medicine, Seoul National University Hospital, Seoul 03080, Korea; collkhk@gmail.com (H.-K.K.); jaho.leigh@gmail.com (J.-H.L.); yeseol.lee127@gmail.com (Y.S.L.); yoonjeong_choi@naver.com (Y.C.); hgseo80@gmail.com (H.G.S.); 2National Traffic Injury Rehabilitation Research Institute, National Traffic Injury Rehabilitation Hospital, Yang-Pyeong 12564, Korea; 3Department of Rehabilitation Medicine, Workers’ Compensation and Welfare Incheon Hospital, Incheon 21417, Korea; 4Department of Health Policy and Management, Seoul National University College of Medicine, Seoul 03080, Korea; yoonkim@snu.ac.kr; 5Institute of Health Policy and Management, Medical Research Center, Seoul National University, Seoul 03080, Korea; 6Department of Neurosurgery, Seoul National University College of Medicine, Seoul 03080, Korea; eunkim@snu.ac.kr (J.E.K.); nsdrcho@gmail.com (W.-S.C.); 7Department of Rehabilitation Medicine, Seoul National University College of Medicine, Seoul 03080, Korea; 8Institute on Aging, Seoul National University, Seoul 03080, Korea; 9Neuroscience Research Institute, Seoul National University College of Medicine, Seoul 03080, Korea

**Keywords:** traumatic brain injury, longitudinal study, epidemiology, incidence, mortality

## Abstract

Traumatic brain injury (TBI), a global public health concern, may lead to death and major disability. While various short-term, small-sample, and cross-sectional studies on TBI have been conducted in South Korea, there is a lack of clarity on the nationwide longitudinal TBI trends in the country. This retrospective study investigated the epidemiological TBI trends in South Korea, using a population-based dataset of the National Health Insurance (2008–2017). The crude and age adjusted TBI incidence and mortality values were calculated and stratified by age, sex, and TBI diagnosis. The age-adjusted incidence per 100,000 people increased until 2010 and showed a decreasing trend (475.8 cases in 2017) thereafter; however, a continuously decreasing age-adjusted mortality trend was observed (42.9 cases in 2008, 11.3 in 2017). The crude incidence rate increased continually in those aged >70 years across all the TBI diagnostic categories. The mortality per 100,000 people was significantly higher among participants aged ≥70 years than in the other age groups. We observed changing trends in the TBI incidence, with a continuously decreasing overall incidence and a rapidly increasing incidence and high mortality values in older adults. Our findings highlight the importance of active TBI prevention in elderly people.

## 1. Introduction

Traumatic brain injury (TBI), which is caused by a strong external impact to the head, has serious implications in terms of long-term physical and psychological health. This type of injury is often referred to as a “silent epidemic” and is a global public health concern, accounting for a large proportion of deaths and disability caused by trauma-related injuries [1].

Approximately 69 million new cases of TBI are reported, worldwide, each year [2]. With some regional differences, recent studies showed a crude incidence rate of 48–849 cases per 100,000 population and a crude mortality of 3.3–28.1 deaths in European countries [3]. Previous studies conducted in the United States, New Zealand, and Europe demonstrated a TBI incidence of 262–798 new cases per 100,000 individuals [3,4,5]. Nevertheless, few studies have been conducted on this subject in Korea, and those previously performed involved limited datasets [6,7,8,9]. Current evidence on TBI has been derived from studies based in the United States, Europe, and Taiwan [10,11,12,13], all of which are characterized by cultures and lifestyles that are in contrast to those in Korea, precluding meaningful comparisons [14].

Accordingly, this study evaluated nationwide TBI incidence and mortality data (2008–2017) from the Korean National Health Insurance Service–National Health Information Database (NHIS-NHID). The purpose of this study was to examine the population-based trends in TBI incidence and mortality in South Korea to support the development of suitable public health policies.

## 2. Materials and Methods

### 2.1. Data Source and Study Population

This retrospective population-based study used data acquired during 2008–2017 by the NHIS-NHID in South Korea. The NHIS is a nationwide, unified, mandatory government-established health insurance system that covers the entire Korean population [15]. In 2018, the total Korean population was 52,556,653, comprising 51,071,982 (97.2%) NHI beneficiaries and 1,484,671 (2.8%) Medical Aid Service recipients [16]. The NHIS-NHID dataset includes information on recipients’ qualifications, premiums, health check-up results, and medical service utilization and healthcare costs; however, it does not include clinical information that can be used for the inference of TBI severity, such as that pertaining to the duration of consciousness loss, post-traumatic amnesia status, and brain imaging findings.

### 2.2. Definition of TBI and Inclusion/Exclusion Criteria

TBI is typically defined, based on criteria specified by the World Health Organization in 2004, as an “acute brain injury resulting from mechanical energy to the head from external physical forces” [17]. The health insurance database in South Korea uses a disease classification system based on the International Classification of Diseases, 10th revision. To investigate the epidemiological patterns of TBI, we extracted data on patients with the following diagnostic codes: Concussion (i.e., S06.0), cranial fracture (i.e., S02.0, S02.1, S02.7, S02.8, and S02.9), or intracranial injury (i.e., S06.1–S06.9). These categories were introduced to enable the assessment of TBI severity. Concussion, in this study, was referred to as a mild TBI, while the last two categories included those with a diagnosis of moderate-to-severe TBI [18]. Patients were included in this study if they were newly diagnosed with TBI at either an inpatient or outpatient department during 2008–2017.

### 2.3. Analysis of Incidence and Mortality

To explore the epidemiological TBI patterns and trends, we conducted descriptive analyses. We used two approaches for the calculation of incidence and mortality values. First, the crude incidence and mortality values per 100,000 persons were calculated using the mid-year population for sex and specific age groups per decade, divided into the following age groups: 0–9, 10–19, 20–29, 30–39, 40–49, 50–59, 60–69, 70–79, and ≥80 years (2008–2017). Deaths that occurred in the same year as the TBI diagnosis were counted. We divided the annual number of new TBI patients and associated deaths by the mid-year population for each year and age group and then multiplied the result by 100,000. The mid-year population was calculated as the average end-of-the-year population recorded over two consecutive years for the minimization of inaccuracies due to recent births and deaths [19]. The annual mid-year population was used to represent the average population, as it is used as a denominator in estimates of birth and death rates, among others [20].

Moreover, we calculated the age-adjusted incidence and mortality values associated with TBI, using the direct standardization method to control for the confounding effect of age. Each adjusted rate represented a weighted average of crude incidence and mortality values per age group, whereby the weights corresponded to the population fraction of persons in a given age group. We used the standard population of South Korea in 2005 as the weight for the calculation of the age-adjusted incidence rate/mortality [21].

### 2.4. Analysis of Odds Ratio and Trends

We conducted trend analysis for the evaluation of the time trends in the crude/age-adjusted incidence and mortality over the study period using the Cochrane-Armitage trend test. Subgroup analyses were performed by age group for the clarification of the increasing and decreasing trends. As we only included deaths that occurred in the same year as the TBI diagnosis, the mortality values may not indicate that the direct cause of death was not TBI. To refine and refill this gap, we calculated odds ratios for the comparison of the mortality values among adults aged ≥70 years with and without TBI in the general population.

Finally, we performed the Cochran-Armitage trend test for the determination of any temporal TBI incidence and mortality trends, including in the age-stratified sub-group analyses. All analyses were performed in SAS version 9.4 (SAS Institute, Cary, NC, USA).

### 2.5. Ethical Approval

The data used in this study were provided by public institutions that collect data at the national level and were used with the requisite permissions. All data were provided after anonymization and retrospectively analyzed. This study was exempted from approval by the institutional review board of Seoul National University Hospital (IRB No. 2003-133-1110). The need for informed consent was waived. This study was performed in accordance with relevant guidelines and regulations.

## 3. Results

### 3.1. TBI Trends in Korea (2008–2017)

Table 1 shows the annual mid-year resident population of Korea during 2008–2017, alongside the number of new TBI patients and the crude incidence rate, by sex. Despite a decreasing trend for both sexes over time, the sex-specific data show that the incidence rate for men was higher from 2008 to 2017 than it was for women (men: 663.6 cases in 2008, 543.9 in 2017, *p* for trend < 0.001, women: 462.3 cases in 2008, 424.4 in 2017, *p* for trend < 0.001).

Figure 1 shows the number of people who used medical services for a diagnosis of TBI between 2008 and 2017. The number of patients who received medical care for TBI increased from 2008 to 2010 (278,288 cases in 2008; 313,455 cases in 2010), and then continuously decreased until 2016 (241,957 cases in 2016). Concurrently, the Korean population showed a continuous increase from 49,054,708 individuals in 2008 to 51,361,911 individuals in 2017; however, there was an absolute frequency decrease of 10.9% in the number of TBI patients in the same period.

### 3.2. Trends in Age-Adjusted TBI Incidence and Mortality (2008–2017)

Figure 2 shows the age adjusted TBI incidence and mortality values in South Korea. A rapid short-term increase was observed in the TBI incidence, from 571.3 cases in 2008 to 638.1 in 2010 (*p* for trend < 0.001), followed by a continuous decrease (475.8 cases in 2017). However, the age adjusted TBI mortality value decreased continuously from 42.9 cases in 2008 to 11.3 in 2017 (*p* for trend < 0.001).

### 3.3. Trend in TBI Incidence by Age and TBI Cause in Korea (2008–2017)

Figure 3 shows the crude incidence rate per 100,000 population divided into 10-year age categories. Participants aged <10 years showed a relatively high incidence of TBI throughout the study period; however, this incidence decreased from 1274.3 cases in 2008 to 998.4 in 2017 (*p* for trend < 0.001). Furthermore, those aged 10–19, 20–29, 30–39, and 40–49 years showed a low average incidence rate, which followed an overall decreasing trend (all *p* for trend < 0.001). Participants aged 50–59 and 60–69 years showed higher TBI incidence rates than their counterparts in the other age groups. A slight decreasing trend in the TBI incidence was noted in the 50–59 years group from 2010 (*p* for trend < 0.01); however, there was no change in the TBI incidence in those aged 60–69 years (*p* for trend = 0.478). In contrast, the 70–79 and ≥80 years age groups showed a steadily increasing TBI incidence trend from 2008 (*p* for trend < 0.01).

Figure 4 shows the age-stratified incidence rate of TBI in two different diagnostic groups: Those with a history of concussion and those with intracranial injuries. The age-adjusted incidence of concussion in 2017 was six times higher (415.2 individuals/100,000 population) than that of intracranial injuries (72.0 individuals/100,000 population). In the concussion group, the crude incidence rate in the 0–9 years group was higher than that in the other age groups, following a decreasing trend (1081.8 individuals in 2008, 921.2 in 2017, *p* for trend < 0.001). Furthermore, although the other age groups showed a slightly decreasing trend in the crude incidence of concussion, the corresponding incidence in those aged ≥80 years showed a steadily increasing trend (509.8 individuals in 2008, 650.7 in 2017, *p* for trend < 0.001).

For intracranial injury, the crude incidence rate in those aged ≥70 years was the highest, showing an increasing trend (216.3 individuals in 2008, 245.6 in 2017, *p* for trend = 0.381). In contrast, the crude incidence rate in patients aged 0–9 years showed a significantly decreasing trend (206.9 individuals in 2008, 88.6 in 2017, *p* for trend < 0.001). During the study period, an increase in the crude incidence of TBI was noted in association with both concussion and intracranial injury among older adults (≥80 years) compared to that in the other age groups. Conversely, in the younger age groups (<40 years), a decreasing TBI incidence trend was observed.

### 3.4. Crude TBI Mortality by Age Group in Korea (2008–2017)

Figure 5 shows the crude mortality values per 100,000 individuals, by age. In general, the mortality values across all the age groups showed a decreasing trend. Particularly, the crude mortality values in the <10, 10–19, 20–29, 30–39, and 40–49 years age groups were relatively low, at 0.7, 1.0, 1.7, 2.1, and 5.2, respectively, in 2017. Concurrently, the mortality values in the 50–59 and 60–69 years age groups were higher than those in the younger groups, showing decreasing trends from 58.8 and 112.2 individuals, in 2008, to 12.6 and 25.6 individuals, in 2017, respectively. Concurrently, the mortality value in the ≥80 years age group was higher than that in the other age groups throughout the study period. However, a gradual decreasing trend was noted from 347.3 individuals in 2008 to 120.6 individuals in 2017. In summary, although the crude mortality values across all the age groups showed a decreasing trend, they also showed a tendency to be higher in the older age groups. When we calculated the odds ratio for the comparison of adults aged ≥70 years with and without TBI, we found: (1) The odds ratio from 2008 to 2017 was greater than 1, and (2) a decreasing trend from 26.9 individuals in 2008 to 4.4 in 2017 (see Appendix A).

## 4. Discussion

### 4.1. Changing Epidemiological Patterns in Korea

The incidence rate of TBI in Korea showed an overall decreasing trend between 2008 and 2017. However, the subgroup analysis showed differences, by age, with decreasing or stable incidence trends observed among patients aged <60 years, compared to the increasing trends noted among those aged ≥70 years. There are several plausible explanations for these age-dependent incidence patterns.

Among older adults (aged ≥70 years), the overall crude incidence rate increased by 24.6% during the study period (Figure 3). The total number of TBI patients increased from 18,510 people in 2008 to 36,271 people in 2017. Similarly, in the TBI diagnosis analysis, the crude incidence rate in the ≥70 years age group increased across the diagnostic categories (Figure 4). Moreover, although the overall mortality decreased, the value in the ≥70 years age group was significantly higher than that in the other age groups (Figure 5). These findings may be related to the rapid increase in the size of the elderly population in South Korea. In the country, the proportion of those aged ≥65 years had increased from 10.2% in 2008 to 14.2% in 2017 [22]. In this study, the average age in the ≥70 years age group had also increased from 77.2 years in 2008 to 78.4 years in 2017. Previous studies reported similar demographic trends and corresponding changes in the TBI incidence among older Korean adults. A longitudinal study of adults aged 65 years or older reported an increase in the overall population size alongside an increase in the number of TBI patients during 1992–2003 [23,24]. Moreover, several studies have reported that falls account for the greatest percentage of TBIs among older adults [25,26].

In patients aged <10 years, a continuous decrease in the crude incidence rate was observed over 10 years, across the TBI diagnostic groups. In particular, this group showed a very high incidence of concussion. Additionally, participants in the youngest age group (0–4 years) showed not only a higher TBI incidence rate, but also a higher TBI mortality rate than in the 5–9 years group. Furthermore, the TBI incidence rate in the 0–4 years group was higher than that in the 0–5 years group in two different diagnostic groups (concussion and intracranial injury; Appendix A). This finding is consistent with that observed in other studies, in which toddlers and young school-aged children who frequently climb, slip, and fall in the playground were found to have an increased risk of concussion, corresponding to a diagnosis of mild TBI [27,28]. In the present study, the incidence rate of concussion in this age group showed a large, continuous decrease over 10 years (2008–2017). This may be related to an increased level of public awareness on the long-term consequences of mild TBI in childhood, which include reduced academic ability, behavioral disorders, and poor general health; in the long term, repeated concussions can cause psychological problems such as anxiety and depression as well as cognitive impairment, dementia, and Parkinson’s disease [29,30,31].

Interest in the consequences of mild TBI in childhood has led to the development of policies and programs aimed at TBI prevention. The United States Centers for Disease Control and Prevention has issued guidelines for the prevention and management of mild TBI in children. Likewise, South Korea has implemented mid-to-long-term research, policies, and intervention agendas for the prevention of safety-related incidents among children; the presently reported decrease in the concussion incidence among those aged <10 years may be a result of these efforts [32]. In addition, as a measure against road traffic accidents, which are major causes of TBI [33], the Korean Ministry of Land, Infrastructure and Transport established a 5-year Basic National Road Traffic Safety Plan, aimed at improving road safety, and reducing the rate of accidents by gaining an understanding of their causes [34]. During the study period (2008–2017), the number of people injured in vehicular accidents in South Korea showed a downward trend, from 692.5 persons/100,000 population in 2008 to 627.5 persons/100,000 population in 2017 [35].

### 4.2. Changing Epidemiological Patterns and Prevention

In this study, we observed an increase in the overall TBI incidence among older adults (age ≥70 years) as well as an increase in the rate of specific TBI diagnoses. In addition, despite a decreasing trend, patients aged 70–79 years and ≥80 years showed a significantly higher mortality value than those in the other age groups and an increasing trend. These findings suggest the need for a TBI prevention policy aimed at older adults; with the increase in the proportion of older adults, the urgency of these policies has become more acute. A previous study showed that older adults are particularly vulnerable to fall-related TBIs; in fact, the associated risk in this age group is six times higher than that in other age groups [36]. Although the causes of TBI were not identified in the present study, the prevention of falls is likely to reduce the incidence of TBI among older adults, among whom they are a major cause of TBI. However, falls have different causes among older adults, including polypharmacy, gait discomfort, living environment, and poor vision [37,38]. Consequently, fall prevention requires research and policy that account for a combination of factors.

### 4.3. Strengths and Limitations

This is the first study to comprehensively investigate the epidemiological trends in TBI in South Korea. While previous studies have reported on TBI incidence in South Korea [39], they included a subset of the population or used estimates from a relatively short period such as a single year. A key strength of the present study is that we used raw nationwide data, allowing for the obtainment of population-based and standardized estimates.

This study has several limitations that should be acknowledged. First, it investigated the general epidemiological patterns of TBI in South Korea; nevertheless, the reported findings are likely to have been underestimated. In South Korea, the NHIS covers 97% of the population; however, data from the Worker’s Compensation Insurance for laborers who experience industrial accidents and those from automobile insurance for people injured in vehicular accidents are managed separately [40]. The rate of trauma related TBI is likely high among people who are injured in industrial or vehicular accidents; however, these cases were not included in this study. Most data on motor insurance in South Korea are managed by individual insurance companies and are inaccessible to the public. However, we were able to inspect some data provided in July 2013 for review by the Health Insurance Review and Assessment (HIRA), which is a public institution. According to the 2017 HIRA report, a total of 79,478 patients used inpatient services for concussion (S06.0), which constitutes a TBI sub-category. However, only data related to concussion were available; therefore, the difference in the incidence rate between insurance services can only be inferred indirectly. As vehicular accidents are usually accompanied by a strong external impact, they are a major cause of TBI [41]. Therefore, to examine the TBI trends in the total population of South Korea, it is paramount that future studies include patients with TBI caused by vehicular and industrial accidents.

Second, we did not evaluate the participants’ TBI diagnosis based on severity; however, we stratified patients into three diagnostic groups. Nonetheless, the same diagnostic code may correspond to different clinical states of varying severities, and there may be differences in the mortality risk and medical utilization patterns based on the presence of concomitant diseases. Thus, future studies are required to investigate the TBI severity patterns and clinical course, including their impact on the epidemiological trends in South Korea.

Third, the reported crude mortality values among patients aged ≥70 years may have been overestimated. In 2008, the incidence rate of TBI among patients aged ≥70 years was 606.0 per 100,000 persons, while the mortality value was 347.3 per 100,000 persons. These results indicate that 57% of those aged ≥70 years who used medical services for TBI died. While NHIS-NHID data can confirm whether or not a patient died, data on the direct cause of death are not available. This suggests that the number of deaths per year included in this study did not represent the deaths caused by TBI; these data represented deaths due to all causes among patients diagnosed with TBI. To control for this dataset-related limitation, at the time of data extraction, we included only deaths that were recorded in the same year as the TBI diagnosis. Additionally, in the calculation of the odds ratio, the mortality values among those aged ≥70 years with TBI was higher than that in the general population aged ≥70 years without TBI. These results suggest that the deaths included in this study were likely caused by TBI.

## 5. Conclusions

This longitudinal study, conducted based on data obtained over a 10-year period, identified the epidemiological trends in TBI in South Korea. Overall, the TBI incidence steadily decreased during the study period; however, it sharply increased among older adults. Similarly, despite an overall decreasing trend in the TBI mortality, the mortality value among adults aged ≥70 years was high. Our results highlight the need for TBI prevention strategies among older adults and may serve as a foundation for further age specific TBI research.

## Figures and Tables

**Figure 1 ijerph-17-06197-f001:**
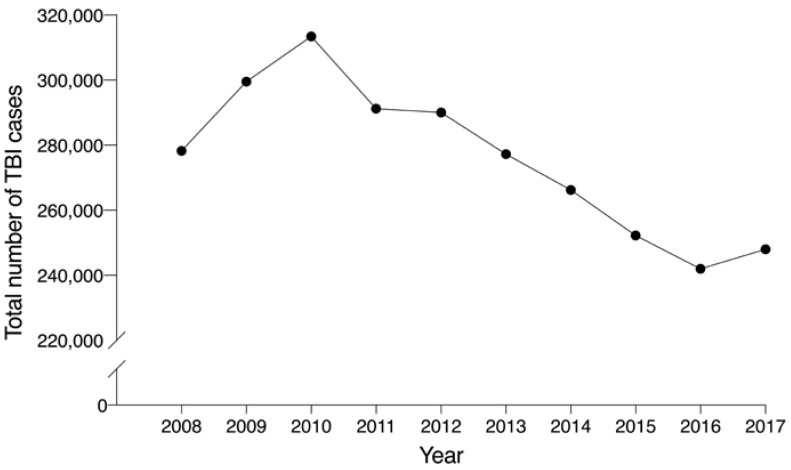
Trends in traumatic brain injury in Korea (2008–2017).

**Figure 2 ijerph-17-06197-f002:**
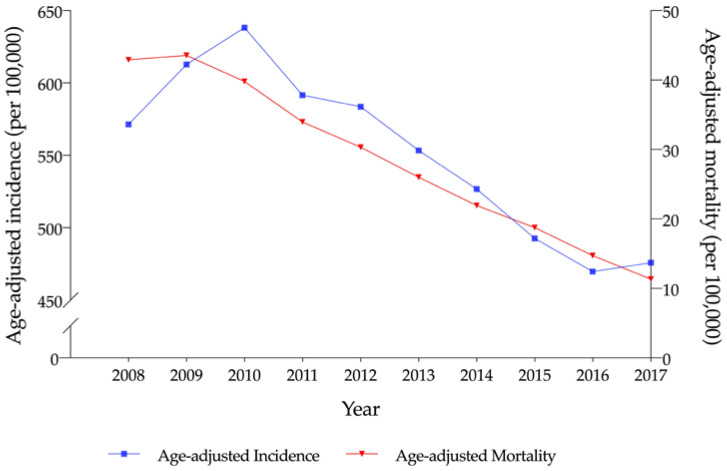
Age-adjusted incidence and mortality values of traumatic brain injury (2008–2017).

**Figure 3 ijerph-17-06197-f003:**
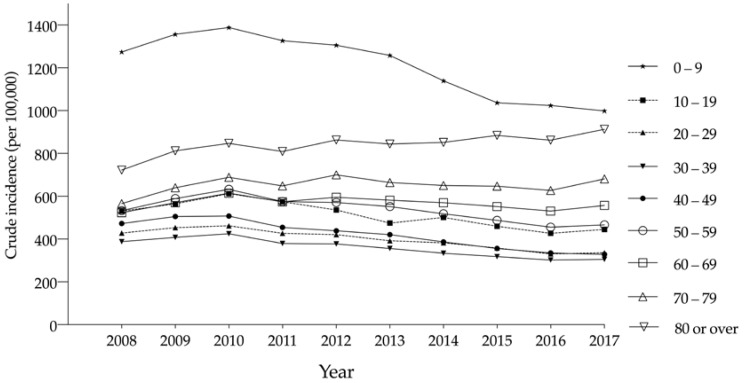
Crude incidence rates (per 100,000) of traumatic brain injury, by age, from 2008 to 2017.

**Figure 4 ijerph-17-06197-f004:**
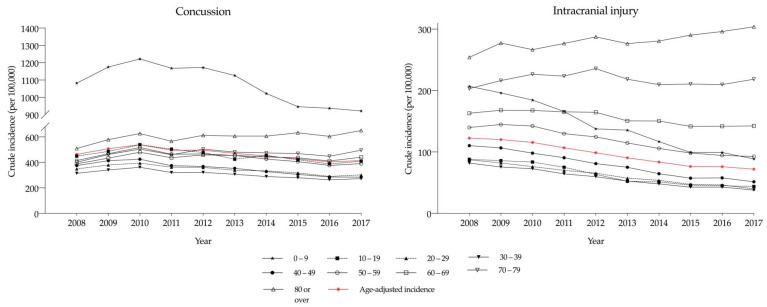
Crude incidence and age-adjusted incidence, by age group, stratified by traumatic brain injury diagnosis.

**Figure 5 ijerph-17-06197-f005:**
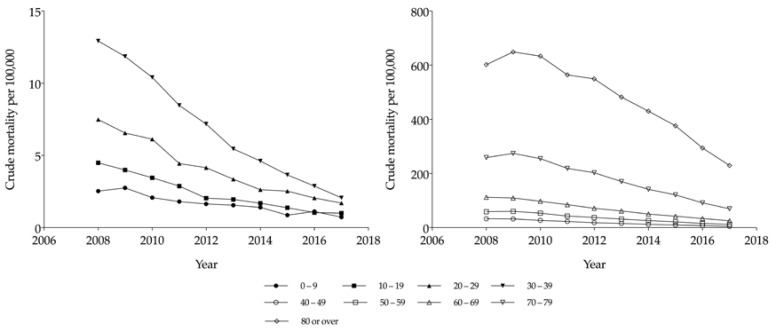
Crude mortality rates (per 100,000 individuals) by age group for the total Korean population (2008–2017).

**Table 1 ijerph-17-06197-t001:** Annual incidence of traumatic brain injury by sex.

	2008	2009	2010	2011	2012	2013	2014	2015	2016	2017
Population										
Men	24,757,073	24,876,418	24,977,163	25,081,787	25,187,494	25,282,928	25,374,486	25,458,057	25,527,814	25,576,752
Women	24,647,574	24,780,338	24,902,648	25,029,688	25,157,830	25,276,023	25,388,672	25,493,661	25,585,157	25,653,952
Total	49,404,647	49,656,756	49,879,811	50,111,475	50,345,324	50,558,951	50,763,158	50,951,718	51,112,971	51,230,704
Number of TBI cases										
Men	164,288	176,584	182,967	169,813	166,984	158,181	151,498	143,598	136,588	139,104
Women	113,940	122,977	130,488	121,395	123,091	119,069	114,734	108,635	105,369	108,885
Total	278,228	299,561	313,455	291,208	290,075	277,250	266,232	252,233	241,957	247,989
Incidence rate										
Men	663.6	709.8	732.5	677.0	662.9	625.6	597.1	564.1	535.1	543.9
Women	462.3	496.3	524.0	485.0	489.3	471.1	451.9	426.1	411.8	424.4
Total	563.2	603.3	628.4	581.1	576.2	548.4	524.5	495.0	473.4	484.1

Note: Incidence per 100,000 individuals. TBI, traumatic brain injury.

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
