# Peer review of "Decreasing Incidence and Mortality in Traumatic Brain Injury in Korea, 2008–2017: A Population-Based Longitudinal Study"

_ijerph, 2020, doi:10.3390/ijerph17176197_

Round 1

Reviewer 1 Report

This is an interesting and well-written paper that could be improved. Here are some questions/comments for the authors:

1) Every half year you calculated the statistics for residents in specific age groups. Why was the mid-year population used as opposed to the year-end? Please justify this and discuss whether/how it impacts the quality of the data or strength of the conclusions.

2) You mention using SAS 9.4 to standardize it to the 2005 data- please provide some details to readers regarding how this was done.

3) I did not notice any explanation of the spike in cases in 2010. Can this be addressed?

4) The most significant limitation is this: Is there any better way to link the death to the TBI or address the fact this cannot be done as a limitation. 

  • If you don’t have the cause of death, what are the odds the 70+ group is just dying because of old age?
  • Cant that be quantified/can the causes of death be obtained on a population-scale and linked to potentially being caused by TBI?

Author Response

Point 1: Every half year you calculated the statistics for residents in specific age groups. Why was the mid-year population used as opposed to the year-end? Please justify this and discuss whether/how it impacts the quality of the data or strength of the conclusions.

Response 1: Thank you for your query. Population estimates at either end of any year tend to be inaccurate due to recent births and deaths; Mid-year population is calculated as the average of end of year population recorded over two consecutive years, not once in the middle of the year. (reference 20). As a result, mid-year population estimate is believed to be a better representation of the population size throughout the year. In fact, the United Nations recommended that vital statistics (i.e. birth rate, death rate, and marriage and divorce rate) be produced using mid-year resident population as a denominator (reference 19). We have added relevant references. According to your recommendation, the meaning of mid-year population has been explained in [page 2 lines 90-93].

 “Mid-year population is calculated as the average of end of year population recorded over two consecutive years to minimize the inaccuracy due to recent births and deaths. Annual mid-year population was used to represent the average population, as it is used as a denominator in estimates of birth and death rates, among others”

Point 2: You mention using SAS 9.4 to standardize it to the 2005 data- please provide some details to readers regarding how this was done.

Response 2: Thank you for your query. We have added a sentence describing the formula of age-adjusted incidence/mortality in the sub section of Materials and Methods. [page 3 lines 94-97]

“Moreover, we calculated age-adjusted incidence and mortality rates associated with TBI, using the direct adjustment standardization method to control for the confounding effect of age. Each adjusted rate represented a weighted average of crude incidence and mortality rates per age group.”

Point 3: I did not notice any explanation of the spike in cases in 2010. Can this be addressed?

Response 3: Thank you for your comment. The TBI case in 2009 and 2010 increased across all age groups compared to the previous year. To explain the spike, we reviewed the literature on changes in national aid related to emergency medical service and changes in resources related to the prehospital emergency transport system. However, we have not yet found adequate literature to explain this spike. The NHIS-NHID data is collected to serve insurance claims; the information it contains can be limited. Should we obtain additional and sufficient data in the future, we will endeavor to examine this question in a separate study.

Point 4: The most significant limitation is this: Is there any better way to link the death to the TBI or address the fact this cannot be done as a limitation.

If you don’t have the cause of death, what are the odds the 70+ group is just dying because of old age?

Can’t that be quantified/can the causes of death be obtained on a population-scale and linked to potentially being caused by TBI?

Response 4: Thank you for bringing this to our attention, which we recognize as a significant limitation of our study. To address it, we calculated mortality odds ratios for adults aged 70+ years and compared them between those with and those without TBI. We

describe the methodology in the Materials and Methods [page 3 lines 103-106],

Because we only included deaths occurred in the same year as TBI diagnosis, mortality rates may not indicate that the direct cause of death was not TBI. To refine and refill this gap, we calculated odds ratios to compare mortality rates among adults aged ≥70 years with and without TBI in general population.

showed specific value of odds ratio in the Results [page 6 lines 182-185]

“When we calculated the odds ratio for comparing adults aged ≥70 years with TBI and without TBI, we found that followings; (1) odds ratio from 2008 to 2017 was greater than one. (2) it shows the decreasing trend from 26.9 in 2008 to 4.4 in 2017 (Supplementary 1).”

and discussed the result in the Discussion [page 8 lines 280-283]

“Also, the result of odds ratio showed that mortality rates of participants aged ≥70 years with TBI was higher than general population aged ≥70 years without TBI. These results suggest that deaths included in this study are likely to be deaths potentially being caused by TBI.”

Reviewer 2 Report

The author have used a national insurance data base to determine the incidence and mortality following ICD 10 diagnosis of head trauma.  they report decreasing incidence of TBI and also decreasing mortality.  Incidence is increasing in the oldest age group as is mortality.   The paper is clearly written, and discussions, while speculative.  Weakness are described

They attribute increasing rates in the elderly to increasing numbers  but since hey are providing rates, this cannot be the explanation.  This is better explained by highest rates in the very old with increasing numbers in the oldest age groups.  I would suggest looking at a finer division of age groups.  Specifically 70-79, and 80+.  A finer age breakdown in those under age 10 would also be useful since factors associated with TBI are quite different in infants.                                    

Author Response

Point 1: I would suggest looking at a finer division of age groups. Specifically, 70-79, and 80+.  A finer age breakdown in those under age 10 would also be useful since factors associated with TBI are quite different in infants.

Response 1: Thank you for this valuable feedback. Following your suggestion, we have separated the older adults age group into 70-79 and 80+ years and compared TBI incidence rates and mortality. As per the reviewer’s recommendation, we have revised and added the results about the figure 3 [page 5 lines 145-146], figure 4 [page 6 lines 155-157] and figure 5 [page 6 lines 161-163] in the manuscript.

[page 5 lines 149-150, figure 3]

“the age group 70-79 and ≥80 years showed a steadily increasing trend in TBI incidence from 2008 (p for trend<0.01).”

[page 5 lines 159-161, figure 4]

“the crude incidence of concussion for the group aged ≥80 years showed a steadily increasing trend (509.8 in 2008; 650.7 in 2017, p for trend<0.001).”

[page 5 lines 165-167, figure 5]

“both the concussion and intracranial injury groups showed an increase in crude incidence of TBI among older adults (80 years), compared to other age groups.”

Reviewer 3 Report

This manuscript is of great interest to the community; however there are major methodological limitations that must be addressed. Specific concerns are below

Methods:

  1. TBI related deaths is a lofty assumption considering that the death is all-cause. Labeling death as TBI-related is a misnomer. A proper definition of deaths is required. The authors can either label deaths as 'deaths' and not 'TBI-related deaths', or the authors can qualify death in some manner. For instance, the authors can examine deaths that occured 30, 60, and or 90 days following date of TBI. Death soon following TBI is more likely to be 'TBI-related'
  2. Define age-adjusted incidence/mortality. The authors need to define how this metric is calculated; how the standard population of South Korea served as the weight.
  3. The analysis section is just one paragraph. This study requires a detailed statistical analysis section. What trend analysis was conducted? What is the level of statistical significance? Without statistical analysis of the data, the trends over time cannot be analyzed; trends are almost meaningless without proper statistical analysis. Consider using Cochran-Armitage trend test or other tests as appropriate.
  4. Define age groups in the methods section rather than results section
  5. Mention TBI severity--was it captured? Consider how severity will impact analysis

Results:

  1. Statements that indicate that trends are decreasing or increasing are inappropriate as no statistical test to examine trends are done
  2. Authors state: "continuously decreasing trend until 2017". This is not true as medical services used for TBI increases from 2016-1017.

Discussion:

  1. It is inaccurate to state that there are decreasing trends, or make any trend statement without a trend analysis
  2. Consider limitations of the data--what is included and what is not?
  3. Consider the limitation of not reporting TBI severity. The authors do distinguish between concussion and intracranial injury, but that does not distinguish between mild, moderate, and severe TBI. Though, concussion is considered mild TBI.

Author Response

Point 1: TBI related deaths is a lofty assumption considering that the death is all-cause. Labeling death as TBI-related is a misnomer. A proper definition of deaths is required. The authors can either label deaths as 'deaths' and not 'TBI-related deaths', or the authors can qualify death in some manner. For instance, the authors can examine deaths that occurred 30, 60, and or 90 days following date of TBI. Death soon following TBI is more likely to be 'TBI-related'

Response 1: Thank you for the valuable comment. when we considered the possible overestimation of crude mortality rates among participants aged ≥ 70 years in the limitation section [page 8 lines 272-283], we described that it did not represent deaths caused by TBI. The responses to the specific comments of the reviewer are as follows:

  • As per the reviewer’s recommendation, we have revised and reworded the term “TBI-related death” as “death” which means all causes deaths to clarify the meaning of deaths in the manuscript. Also, we have redefined the death which occurred in the same year as TBI diagnosis in the method section [page 2 lines 87-88

Method section [page 2 lines 87-88]

“Deaths that occurred in the same year as TBI diagnosis were counted.”

  • To find better way to link the death to the TBI, we calculated odds ratio to compare death rates among participants aged ≥70 years with and without TBI. We have included sentences describing the relationship between deaths due to all causes and TBI in the statistical section [page 3 lines 103-106], result section [page 6 lines 182-184] and discussion section [page 8 lines 280-283].

Statistical section [page 3 lines 103-106]

“Because we only included deaths occurred in the same year as TBI diagnosis, mortality rates may not indicate that the direct cause of death was not TBI. To refine and refill this gap, we calculated odds ratios to compare mortality rates among adults aged ≥70 years with and without TBI in general population.”

Result section [page 6 lines 182-184]

“When we calculated the odds ratio for comparing adults aged ≥70 years with TBI and without TBI, we found that followings; (1) odds ratio from 2008 to 2017 was greater than one. (2) it shows the decreasing trend from 26.9 in 2008 to 4.4 in 2017 (Supplementary 1). These results suggest that deaths included in this study are likely to be deaths potentially being caused by TBI.”

Discussion section [page 8 lines 280-283]

“Also, the result of odds ratio showed that mortality rates of participants aged ≥70 years with TBI was higher than general population aged ≥70 years without TBI.”

Point 2: Define age-adjusted incidence/mortality. The authors need to define how this metric is calculated; how the standard population of South Korea served as the weight.

Response 2: Thank you for bringing this aspect of our methods to our attention, as it needs additional explanation. We have revised the methods section to include details of standardization based on age distribution and corresponding weights, derived from the 2005 standard population of Korea and have added relevant references about standard population (reference 21). [page 2, lines 94-97]

“Moreover, we calculated age-adjusted incidence and mortality rates associated with TBI, using the direct adjustment standardization method to control for the confounding effect of age. Each adjusted rate represented a weighted average of crude incidence and mortality rates per age group, whereby the weights corresponded to the population fraction of persons in a given age group.”

Point 3: The analysis section is just one paragraph. This study requires a detailed statistical analysis section. What trend analysis was conducted? What is the level of statistical significance? Without statistical analysis of the data, the trends over time cannot be analyzed; trends are almost meaningless without proper statistical analysis. Consider using Cochran-Armitage trend test or other tests as appropriate.

Response 3: Thank you for bringing to our attention the insufficient information provided in the analysis section. Accordingly, we have provided the required details. Per your suggestion, we performed the Cochran-Armitage trend test and reported p-values for all trend analysis results.

We described the Cochrane-Armitage trend test in [page 3, lines 101-103].

We conducted trend analysis to evaluate time trends of the crude/age-adjusted incidence and mortality over the study’s time period using the Cochrane-Armitage trend test. Subgroup analysis was performed by age group to clarify the increasing and decreasing trends.

Also, the p-value for trend has described in the result section as follows;

“p for trend<.001” or “all p for trend<0.001”

Point 4: Define age groups in the methods section rather than results section

Response 4: Thank you for your comment. I apologize for the confusion, and have revised the sentence which now reads “…by using the resident mid-year population for specific age groups per decade, divided into the following age groups: 10–19, 20–29, 30–39, 40-49, 50-59, 60-69, 70-79 and ≥80 years (2008 -2017).” [page 2 lines 86-88]

Point 5: Mention TBI severity --was it captured? Consider how severity will impact analysis

Response 5: Thank you for your comment. Due to the limitations of our data source, we were not able to obtain or infer information on TBI severity, including duration of loss of consciousness, post-traumatic amnesia status, or brain imaging findings. To account for different degrees of severity, we have categorized TBI types into three groups, based on the ICD-10 diagnostic codes; TBI severity was inferred accordingly. We have revised our methods section to include all relevant details of this aspect of our study in page 2, lines 78-74.

“…with the following diagnostic codes: concussion (i.e., S06.0), cranial fracture (i.e., S02.0, S02.1, S02.7, S02.8, and S02.9), or intracranial injury (i.e., S06.1–S06.9). These categories were introduced to enable assessment of TBI severity; we refer here to concussion as mild TBI and last two categories of diagnosis as moderate to severe TBI.”

Results:

Point 6: Statements that indicate that trends are decreasing or increasing are inappropriate as no statistical test to examine trends are done

Response 6: The authors would like to thank the reviewer for the constructive critique of the manuscript. The concern regarding the statistical test for trends has been addressed in detail in our response to Reviewer 3, Point 3 above. Briefly, the authors have added the result of trend test, Cochran-Armitage trend test in the Materials and Methods and Results.

[page 3 lines 101-103]

We conducted trend analysis to evaluate time trends of the crude/age-adjusted incidence and mortality over the study’s time period using the Cochrane-Armitage trend test. Subgroup analysis was performed by age group to clarify the increasing and decreasing trends.

[page 4-5]

“p for trend<.001” or “all p for trend<0.001”

Point 7: Authors state: "continuously decreasing trend until 2017 ". This is not true as medical services used for TBI increases from 2016-2017.

Response 7: Thank you for pointing out the inaccuracy sentence. We apologize for the confusion and inaccuracy, and have revised the sentence based on the results which now reads “…and then a continuously decreasing trend until 2016 (241,957 in 2016).”
[page 4 lines 127-128]

Discussion:

Point 8: It is inaccurate to state that there are decreasing trends, or make any trend statement without a trend analysis

Response 8: Thank you for your comment. We agree with the reviewer’s comment that the lack of statement for trends such as increasing and decreasing trends is a significant weakness to present our results. As we respond to Reviewer’s Point 3 and Point 6 above, we performed the Cochran-Armitage trend test to state that there are increasing or decreasing trends. [page 3 lines 107-108]

“Finally, we performed the Cochrane-Armitage trend test to determine any temporal TBI incidence and mortality trends, including in age-stratified sub-group analysis.”

[page 4-5]

“p for trend<.001” or “all p for trend<0.001”

Point 9: Consider limitations of the data--what is included and what is not?

Response 9: Thank you for your useful suggestions. The clinical measures including loss of consciousness duration, post-traumatic amnesia status, or brain imaging findings were not collected in NHIS-NHID we used. In this reason, we restrictively divided the TBI into diagnostic groups; as inferred the severity of TBI. As per the reviewer’s recommendation, we have added sentences describing the limitation of the data;

(1) in the Materials and Methods [Page 2 lines 68-70],

“however, it does not include clinical information to infer TBI severity, such as loss of consciousness duration, post-traumatic amnesia status, or brain imaging findings.”

(2) in the Discussion [Page 8 lines 267-269]

“Nonetheless, the same diagnostic code might correspond to different clinical states of varied severity, and there may be differences in mortality risk and medical utilization patterns based on concomitant diseases”

Point 10: Consider the limitation of not reporting TBI severity. The authors do distinguish between concussion and intracranial injury, but that does not distinguish between mild, moderate, and severe TBI. Though, concussion is considered mild TBI.

Response 10: Thank you for your constructive comment. In response to your feedback, we have addressed the issue of TBI severity in our manuscript, where we removed that we refer to concussion as mild TBI. Despite of the restriction of the data, we tried to infer the severity of TBI to categorize TBI types into three groups, based on ICD-10 diagnostic codes. We have revised sentence in the method section [page 2 lines 76-79], described in the limitation section [page 8 lines 266-269].

Method section [page 2 lines 76-79]

“…with the following diagnostic codes: concussion (i.e., S06.0), cranial fracture (i.e., S02.0, S02.1, S02.7, S02.8, and S02.9), or intracranial injury (i.e., S06.1–S06.9). These categories were introduced to enable assessment of TBI severity; we refer here to concussion as mild TBI and last two categories of diagnosis as moderate to severe TBI [18].”

Discussion section [page 8 lines 266-269]

 “Second, we did not evaluate TBI diagnosis based on severity; however, we stratified patients into three diagnostic groups. Nonetheless, the same diagnostic code might correspond to different clinical states of varied severity, and there may be differences in mortality risk and medical utilization patterns based on concomitant diseases.” [page 8 lines 249-252]

Round 2

Reviewer 2 Report

Need to add 0-9 in methods

Supplementary table should be beter labeded.  Wht is the comparison?

Woul still like to see 0-4, 5-9 age groups. 

Reviewer 3 Report

Thank you for adequately addressing comments. This is a very strong and important manuscripts. I especially appreciate your addition of odds ratios. I think it is almost ready for publication. There are a few minor comments regarding the results:

  1. Page 3, line 121-123, the authors state: "...despite a decreasing trend over time for both sexes...", please follow this up with a p-value. Is this trend significant?
  2. Page 4, lines 125-130: I would avoid using the word 'trend' (i.e. 'increasing trend', 'decreasing trend') if significance is not mentioned. Either add p-value or just indicate that patients who received medical care decreased/increased.
